# Cationic Microbubbles for Non-Selective Binding of Cavitation Nuclei to Bacterial Biofilms

**DOI:** 10.3390/pharmaceutics15051495

**Published:** 2023-05-13

**Authors:** Gareth LuTheryn, Elaine M. L. Ho, Victor Choi, Dario Carugo

**Affiliations:** 1Nuffield Department of Orthopaedics, Rheumatology and Musculoskeletal Sciences (NDORMS), The Botnar Research Centre, University of Oxford, Windmill Road, Oxford OX3 7HE, UK; 2Faculty of Engineering and Physical Sciences, University of Southampton, University Road, Southampton SO17 1BJ, UK; elaine.ho@rfi.ac.uk; 3Artificial Intelligence and Informatics, The Rosalind Franklin Institute, Harwell Campus, Didcot OX11 0QX, UK; 4Institute of Biomedical Engineering, Department of Engineering Science, University of Oxford, Parks Road, Oxford OX1 3PJ, UK; victor.choi@eng.ox.ac.uk

**Keywords:** microbubble, cavitation nuclei, Biofilm, microbubble targeting, cationic microbubble, drug delivery, ultrasound drug delivery

## Abstract

The presence of multi-drug resistant biofilms in chronic, persistent infections is a major barrier to successful clinical outcomes of therapy. The production of an extracellular matrix is a characteristic of the biofilm phenotype, intrinsically linked to antimicrobial tolerance. The heterogeneity of the extracellular matrix makes it highly dynamic, with substantial differences in composition between biofilms, even in the same species. This variability poses a major challenge in targeting drug delivery systems to biofilms, as there are few elements both suitably conserved and widely expressed across multiple species. However, the presence of extracellular DNA within the extracellular matrix is ubiquitous across species, which alongside bacterial cell components, gives the biofilm its net negative charge. This research aims to develop a means of targeting biofilms to enhance drug delivery by developing a cationic gas-filled microbubble that non-selectively targets the negatively charged biofilm. Cationic and uncharged microbubbles loaded with different gases were formulated and tested to determine their stability, ability to bind to negatively charged artificial substrates, binding strength, and, subsequently, their ability to adhere to biofilms. It was shown that compared to their uncharged counterparts, cationic microbubbles facilitated a significant increase in the number of microbubbles that could both bind and sustain their interaction with biofilms. This work is the first to demonstrate the utility of charged microbubbles for the non-selective targeting of bacterial biofilms, which could be used to significantly enhance stimuli-mediated drug delivery to the bacterial biofilm.

## 1. Introduction

Biofilms present a multifaceted challenge to achieving a positive therapeutic outcome in chronic infections, principally due to their competency as a physicochemical barrier to treatment [1,2]. Biofilms are typically implicated in severe chronic infection and disease such as in chronic wounds and cystic fibrosis, which represent an estimated global annual expenditure of $281 and $7.5 billion, respectively [3]. Moreover, biofilms represent the world’s leading cause of antimicrobial resistance, which a UK report anticipated would lead to 10 million deaths worldwide by 2050 and a $100 trillion burden to the global economy [4]. For these reasons, research pertaining to the development of novel antibiofilm technologies and pharmaceutical agents has gained substantial momentum. Growing research suggests that targeting the delivery of antimicrobial agents to biofilms may increase their therapeutic potential [5,6,7] by increasing their local concentration to overcome the biofilm’s innate tolerance and reducing the negative systemic side-effects of antimicrobial therapy [8,9]. To achieve greater control over drug delivery and enhance therapeutic efficacy, the use of ultrasound responsive agents has rapidly become an important area of research. Microbubbles (MBs) are small, gas-filled and shelled spherical particles, typically between 0.5 µm and 10 µm in diameter [10,11]. The composition, application, and versatility of MBs, as both ultrasound contrast agents in diagnostics and cavitation nuclei for drug delivery, have been comprehensively reviewed elsewhere [11,12,13,14,15,16,17,18]. Briefly, the composition of a microbubble shell is a key factor in determining the MB stability, biophysical effects in response to ultrasound exposure, and therapeutic outcomes. Enhancing the localisation of MBs to the target treatment site is acutely linked to their therapeutic efficacy. Some strategies employed for targeted drug delivery and gene therapy with functionalised MBs rely on the use of biochemical ligand-receptor targeting, acoustic radiation forces, electrostatic charge, and magnetic manipulation [19,20,21,22]. 

Due to the heterogeneity of the extracellular polymeric substances (EPS) that constitute the extracellular matrix of the biofilm, the identification of an appropriate, commonly expressed target receptor has proved challenging [7]. Recent work has demonstrated that microbubbles functionalised with an Affimer protein can be utilised to selectively target the *Staphylococcus aureus* biofilm virulence factor, clumping factor A [23]. One potential ligand target present in the extracellular matrix of clinically significant biofilms, such as *Pseudomonas aeruginosa*, are lectins [24]; they are carbohydrate-binding proteins that play an important role in mediating initial bacterial adhesion to surfaces for biofilm formation and subsequent polysaccharide cross-linking in the extracellular matrix of biofilms [25]. A significant limitation of this method of biofilm targeting, however, is that lectin receptors have a very weak natural affinity for their carbohydrate ligands, making them a difficult target in an already impervious structure [6,26]. The affinity that a ligand has for the intended receptor also suffers from high variability across biological systems; this is a widespread issue for ligand-receptor mediated specific targeting [27]. Importantly, authors have previously outlined that such means of targeting may not be directly translatable to other strains of the same bacterial species [6,23,25]. This naturally raises the issue of how successfully this means of targeting can be implemented clinically, where multi-species biofilms are abundant and specific strains often unidentified. Due to the challenges in current research on specific ligand-receptor mediated targeting of bacterial biofilms, it was the aim of this research to validate the use of a non-selective means of targeting the bacterial biofilm. Regardless of constituent species or specific strains, bacterial biofilms exhibit a net negative charge. This is owed to the ubiquitous presence of negatively charged extracellular DNA and polysaccharide constituents of the biofilm extracellular matrix, as well as highly negatively charged bacterial cell membrane components such as teichoic acids and lipopolysaccharides [28,29]. It has been demonstrated for example, that *P. aeruginosa* biofilms have a zeta potential (i.e., a measure of the electrostatic charge at the biofilm’s surface) in the range of −20 mV [30].

The incorporation of cationic lipids allows MBs to electrostatically interact with negatively charged molecules; this method has already been used to bind nucleic acids to MBs for gene delivery [31,32]. An interesting point to note is that MBs augmented with a charge have not been used as an explicit means to facilitate MB interaction with biofilms. Recent work in this area has shown that cationic microbubbles can selectively localise in tumour vasculature [33], which demonstrates the utility of electrostatic interactions as a means of targeting regardless of the specific biomarkers present. In applications requiring intravascular administration of MBs, non-specific interactions of MBs with endothelial cells or blood constituents before reaching their target site would however be undesirable. Therefore, the use of cationic MBs for non-selectively targeting of specific sites or features is not well explored. For non-intravenous therapies however, such as the treatment of superficial wound infections, utilising positively charged MBs can provide the means to induce a prolonged MB-biofilm interaction unhindered. This application has been explored in conjunction with the delivery of therapeutic gases such as nitric oxide (NO), which has shown to induce dispersal in *P. aeruginosa* biofilms [34,35]. The use of a non-selectively targeted MB has demonstrated improved treatment efficacy by reducing the diffusion distance between this highly reactive gas and the biofilm. While utility and success of this cationic MB formulation and therapeutic approach have been demonstrated in other work [36], the present study aims to outline the formulation and testing underpinning its successful implementation. Herein, we hypothesised that cationic microbubbles could therefore be utilised as a non-selective means of targeting bacterial biofilms to achieve greater local concentration of MBs associated with and proximal to the biofilm and in turn increase the efficacy of ultrasound-mediated antibiofilm therapy. 

## 2. Materials and Methods

### 2.1. Production of MBs

Room air microbubbles (RAMBs) and NO-filled MBs (NOMBs) were produced as previously described [36,37]. Briefly, 1,2-distearoylphosphatidylcholine (DSPC) (850365P, Avanti, Sigma-Aldrich, Alabaster, AL, USA) and polyoxyethylene (40) stearate (PEG40s) (P3440, Sigma-Aldrich, St. Loui, MO, USA) dissolved in chloroform were combined in a 20 mL capacity and 23 mm inner diameter glass vial (15394769, Fisherbrand™, Fisher Scientific, Dublin, Ireland) in a 9:1 molar ratio, using a 1 mL Luer lock glass syringe (1MR-GT, S.G.E Gas Tight Syringe, Supelco, UK) to achieve a final lipid concentration of 4 mg/mL. The chloroform was evaporated to obtain a dry lipid film; this was then rehydrated with 5 mL of degassed 0.01 M sterile phosphate-buffered saline (PBS) (P4417, Sigma-Aldrich, Gillingham, UK). Each vial was placed on a stirring hotplate (Fisherbrand™, Isotemp™) for 30 min at a temperature of 90 °C and at 700 rpm. Using a 120 W, 3.175 mm diameter tip sonicator (20 kHz, Fisher Scientific FB120, Pittsburgh, PA, USA), the lipid suspension was homogenously dispersed for 150 s at 40% power (48 W), with the sonicator tip fully immersed in the liquid. Microbubbles were subsequently formed by placing the sonicator tip at the liquid–air interface of the homogenised lipid suspension for 30 s at 70% power (84 W). Upon completion of the second sonication step, the vial was placed immediately into an ice bath to rapidly cool the MB suspension. 

For the generation of nitric oxide microbubbles, the same sonication steps were performed; however, the dry lipid film was reconstituted with PBS that was sparged with pure nitrogen for 20 min [38]. Thereafter, a constant flow of nitrogen was used to purge air from the headspace of the vial. The eNO generator (NitricGen Inc., Madison, WI, USA) was used to flush 40 ppm NO at 1.5 L/min through the sparged PBS during the first sonication step [39]. During the second sonication step, the flow of NO was maintained with the needle placed at the interface between the fluid and the sonicator tip. 

Cationic MBs (+) were produced by adding the cationic phospholipid 1,2-distearoyl-sn-glycero-3-ethylphosphocholine (DSEPC) (Avanti Polar Lipids, Alabaster, AL, USA) dissolved in chloroform to the DSPC and PEG40s mixture in quantities determined by the desired molar ratio, final lipid concentration, and sample volume as described in other research [36,37,40,41] (see Table 1).

### 2.2. Development of Microfluidic Flow Cells for Creating Surfaces with Augmented Electrostatic Charge

Flow cells were designed using Solidworks (Solidworks 2018, Dassault Systèmes, France) and converted to DXF files, which were compatible with the software of a laser cutting machine (Lasercut 5.3, HPC Laser Ltd., Yorkshire, UK). The flow cells were designed to fit a 75 mm × 25 mm glass slide. The inlet and outlet ports were laser cut (LS1690, HPC Laser Ltd., Yorkshire, UK) from a layer of 600 mm thick clear acrylic. Double-sided tape (0.14 mm thick, 3M™ Double Coated Polyester Tape 9731, Self-Adhesive Supplies, Reading, UK) was laid over the acrylic, and the channels were laser cut from the tape. A glass substrate was then fixed over the double-sided tape (Figure 1). Untreated borosilicate glass coverslips (0CON-161, 75 × 25 × 0.17 mm glass coverslips, Logitech, Glasgow, UK) were used as an uncharged substrate in the flow cell. Quartz coverslips (Alfa Aesar™ Quartz Microscope Slides, Fisher Scientific UK Ltd.) were instead used for creating flow cells with a substrate having a negative electrostatic charge, as quartz becomes relatively anionic when in contact with a solution (e.g., PBS) at pH 7.4 [42]. To ensure the flow cell was fluid-tight, water containing a red food dye was pumped through the flow cell at increasing flow rates until failure (i.e., corresponding to visible leakage) occurred. Two parallel channels were used to increase the throughput of the device and allow comparative studies to be performed with biofilms grown under the same conditions. A high aspect-ratio design was used for the channel cross-section to provide relatively uniform wall shear stress over the substrate surface. 

### 2.3. Assessment of MB Charge

The charge associated with each MB shell composition was assessed using a dynamic light scattering apparatus (DLS; Zetasizer Nano ZS, Malvern Instruments, Malvern, UK); MBs were prepared as outlined in the above protocol except for the second sonication step. The lipid vesicles obtained from this process were diluted immediately prior to measurement (30 µL in 970 µL) in 10 mM HEPES buffer. Samples were centrifuged at 10,000 rpm for 30 s to remove any formed bubbles which would not be compatible with the measurement technique. Measurements were performed using the Smoluchowski algorithm for up to 100 runs, with three measurements conducted per sample. 

### 2.4. Determination of MB Binding Characteristics under Static Conditions

To determine if the incorporation of a positive charge into the MB shell can promote contact with negatively charged surfaces, an artificially charged environment was created for controlled preliminary testing. Firstly, a flow cell with an uncharged glass coverslip was primed with PBS. Tubing (4 mm inner diameter, Masterflex Transfer Tubing, Platinum-Cured Silicone, WZ-95802-03, Cole Parmer, Saint Neots, UK) was fitted to the inlet and outlet of the flow cell. Approximately 1 mL of each undiluted MB suspension was manually injected into the flow cell via the tubing until the suspension was visible in the outlet tubing. To prevent back flow of the MB suspension, the outlet tubing was then clamped to create a seal. A total of 10 ‘before interaction’ images were acquired at 50 mm intervals from the inlet (left) to the outlet hole (right), along the centreline of the flow cell channel. Images were captured under bright field microscopy (Olympus, IX71), with a 50× (Olympus, LMPLFLN) objective lens using a CCD camera (Hamamatsu ORCA-ER, C4742-80) and analysed in accordance with Appendix A. The centreline of the channel was approximately 1.7 mm equidistant from the channel side walls, as indicated in Figure 1, and was selected for MB imaging purposes, as the wall shear stress acting over this region of the glass surface would be relatively uniform. Each MB suspension was allowed to interact with the charged or uncharged surface for 60 s; this interaction was facilitated by reversing the orientation of the flow cell. This meant the neutral or charged surface was positioned to be the ‘top’ of the flow cell, allowing MBs to passively float towards and interact with the surface. After 60 s, the device was reoriented so the neutral or charged surface was again positioned at the bottom of the flow cell; a further 10 ‘post interaction’ images were then captured and analysed in the same way as previously described. A 60 s interaction period was selected as a representative clinically viable treatment timescale for a topical administration; this also aligns with the time taken for the majority of the MBs in suspension to passively float up towards the surface. This procedure was repeated for three independent vials of each MB composition with an uncharged and charged surface. The percentage change in MB interaction with each surface was calculated using Equation (1); where *V*_1_ corresponds to the MB concentration (in MB/mL) present before contact with the charged or uncharged surface was induced (i.e., glass surface facing upwards), and *V*_2_ corresponds to the concentration of MBs that remained in contact with the charged or uncharged surface after induced interaction was stopped (i.e., glass surface facing downwards). 

Equation (1): Percentage change in MB concentration.
(1)$$\left[\begin{array}{c} Change\;in\\ MB\;concentration\;(\%) \end{array}\right]=\frac{V_{2}-V_{1}}{V_{1}}\times 100$$

### 2.5. Assessment of MB Non-Selective Binding Strength under Flow

The mean wall shear stress exerted by the fluid flow over the channel substrate was varied by adjusting the flow rate of fluid through the channel. The volumetric flow rate required to generate a given mean wall shear stress was calculated by first determining the pressure drop along the channel length (for a given wall shear stress value). The pressure drop was calculated from Equation (2) [43], where ∆*P* = pressure drop along the channel (Pa), *τ* = mean wall shear stress (Pa), *P_W_* = wetted channel perimeter (m) = 2 × channel width + 2 × channel depth (assuming the channel was completely filled, *P_W_* = 0.00728 m), *L* = channel length (m) = 0.0655 m, and *A* = cross-sectional area of the channel (m^2^) = 4.9 × 10^−7^ m^2^.

Equation (2): Pressure drop along the flow cell channel.
(2)$$∆P=\frac{\tau \times P_{w}\times L}{A}$$

The required volumetric flow rate was subsequently calculated from the pressure drop along the channel using Hagen–Poiseuille’s law, which was corrected for calculating volumetric flow rate within a rectangular channel [44], where *Q* = volumetric flow rate (mL/min), ∆*P* = pressure drop along the channel length (Pa), *D_H_* = channel hydraulic diameter (m) = 4 × cross-sectional area / wetted perimeter = 2.7 × 10^−4^ m, *µ* = dynamic viscosity of water = 0.00089 kg/ms, *L* = channel length (m), and 0.2 is the correction factor for a rectangular channel with a width of 3.5 mm and depth of 0.14 mm. The calculated volumetric flow rates required to achieve mean wall shear stress values of 0.1, 0.2, 0.5, and 1 Pa are shown in Table 2.

Equation (3): Hagen–Poiseuille’s law corrected for estimating flow rate in a rectangular channel.
(3)$$Q=\frac{∆P\times \pi \times D_{H}^{4}}{128\times \mu \times L\times 0.2}\times \frac{10^{6}}{60}$$

The MB concentration was assessed after increasing the mean wall shear stress acting on the glass substrate to determine the MB binding strength to uncharged and anionic substrates. The MB suspension was drawn into a 5 mL syringe (BD Luer Lock, 309649) and manipulated by inversion to ensure homogeneous distribution of MBs. A second 10 mL Luer Lock syringe (BD Luer Lock, SYR912) was filled with sterile 0.01 M PBS. Both syringes were connected to the inlets of a three-way valve (Masterflex Fitting, polycarbonate, Three-Way, Stopcock with Male Luer Lock, UY-30600-02, Cole Parmer, UK), with the outlet connected to the flow cell through a segment of tubing. The tubing and flow cell were primed with PBS before the MB suspension was pumped into the flow cell at a flow rate of 0.01 mL/min until the channel was filled. When the channel was full of MB suspension in a 1:1 volumetric ratio with PBS, the flow cell was placed with the glass surface facing up for 60 s to allow interaction of the MBs with the uncharged/charged surface. PBS was then pumped through the channel at 0.01 mL/min for 90 s to remove any unbound or weakly associated MBs. 

After visual inspection confirmed MBs were stationary within the channel, ten images were acquired at 5 mm intervals along the centreline of the channel. This ensured the full length of the channel occupied by MBs was analysed, with a direct comparison of the same areas of the channel before and after application of flow. PBS was pumped into the flow cell for 15 s at a time, with flow rates of 0.06, 0.13, 0.32, and 0.65 mL/min respectively, to achieve wall shear stress values shown in Table 2. After the application of each flow rate, ten images at the same 5 mm positions along the centreline of the channel were acquired after visual confirmation that MBs had become stationary. The procedure was repeated in triplicate with both MB compositions in uncharged and charged flow cells. Segmentation of individual MBs was not possible for this experiment as there was significant clustering of the MBs. Therefore, in an adaptation of the method outlined in Appendix A, the total percentage area covered by the MBs in each image was determined by image thresholding and binarisation in ImageJ (NIH). The total area covered by MBs obtained through thresholding was calculated for each set of 10 images at each flow rate and for each MB composition. For each formulation, the mean area covered by MBs derived from 10 images at each flow rate was normalised by dividing the mean area by the initial area covered by MBs. This is represented by Equation (4); where $\overset{-}{x}$ corresponds to the mean percentage area of MBs present in the channel after flow rates of 0.06, 0.13, 0.32, or 0.65 mL/min, respectively, were applied, and the mean percentage area of MBs present in the channel prior to the application of flow is denoted as *µ*. The results of Equation (4) equate the initial percentage coverage for each formulation to 100%, whilst providing the proportional change in MBs due to the application of various wall shear stress values. 

Equation (4): Normalisation of percentage area covered by MBs.
(4)$$\left[Normalised\;MB\;Area\left(\%\right)\right]=\frac{\overset{-}{x}}{\mu }\times 100$$

### 2.6. Assessment of Non-Selective Interaction of MBs with P. aeruginosa Biofilms

*P. aeruginosa* (PAO1) biofilms were grown in Ibidi^®^ dishes (µ-Dish 35 mm, glass bottom, Thistle Scientific) for 24 h in wound constituent medium (WCM) at 37 °C as described previously [36]. Biofilms were washed three times with sterile PBS to remove any planktonic or weakly attached cells from the dish growth area. Biofilms were live/dead stained with 2.5 µM Syto9 (S34854, Invitrogen™, ThermoFisher Scientific) and 9 µM propidium iodide (P3566, Invitrogen™, ThermoFisher Scientific) for 5 min and stored under foil to prevent exposure to light. Excess stain was removed by washing with sterile PBS and visual confirmation of biofilm’s presence was performed with fluorescence microscopy before the application of MBs (using an EVOS M5000 optical microscope). 

Cationic DSPC RAMB^+^ and NOMB^+^ and uncharged DSPC RAMB and NOMB suspensions were diluted 1:5 (by volume) in sterile PBS and 1 mL of MB suspension was gently pipetted onto the previously live/dead stained biofilm. Each MB formulation was tested in triplicate using three independent biofilm samples (n = 18); the total MB-biofilm interaction time was 60 s for each biofilm. After 60 s, the Ibidi^®^ dish was held at a 45° angle and washed from the top with 1 mL of sterile PBS three times; this ensured removal of MBs not interacting strongly with the biofilm. Each biofilm was imaged with the EVOSM5000 microscope (using a 20× Plan Fluor EVOS AMEP42924 objective); green fluorescent Syto9 was assessed under light emission at 510 nm, gain of 5, light source intensity of 0.5, brightness of 0.5, exposure time of 20 ms, and using a green fluorescent protein (GFP) LED cube (AMEP4653, Invitrogen™, ThermoFisher Scientific). Red fluorescent propidium iodide was assessed under light emission at 593 nm, gain of 5, light source intensity of 0.5, brightness of 0.5, exposure time of 20 ms, 20× magnification objective, and using a red fluorescent protein (RFP) LED cube. MBs were observed in brightfield to assess their localisation pattern, and biofilm association was calculated as percentage area using ImageJ in an adaptation of the steps outlined in Appendix A. Briefly, the brightfield images of MBs were isolated from the combined biofilm-MB images. An automatic threshold was applied to the image before a mask of each MB was created to eliminate the gas core. Based on the total area of the image, the percentage area occupied by MBs was then calculated by the software.

### 2.7. Statistical Analysis 

All data were assessed for normal distribution. For normally distributed data, both paired and independent t-tests have been used to compare data depending on the relationship between variables. For multiple treatment conditions, a one-way ANOVA was used to identify any significant difference between groups. All data were analysed and plotted using Prism 8.4.3 (GraphPad), with a threshold value for significance of <0.05 where * = *p* < 0.05, ** = *p* < 0.005, *** = *p* < 0.0005, and **** = *p* < 0.0001.

## 3. Results

The mean size, concentration, and zeta potential of each MB formulation immediately after production are shown in Table 3; the results are averaged from three replicates with standard deviation of the mean shown where applicable. There was no statistically significant difference between the size or concentration of each formulation of MBs, consistent with previous work on neutral and cationic MBs [45]. Full data are shown in the Appendix A (Figure A2).

### 3.1. Microbubble Binding Characteristics in Static Conditions

In testing which involved no flow of fluid through the channel after MB administration, cationic room air MBs (RAMBs^+^) showed a significant (*p* < 0.05) increase in surface interaction compared to their uncharged counterparts (Figure 2). There was approximately a 40% increase in surface-associated MBs. The results for room air MBs (RAMBs) demonstrate that when the charged lipid is removed, the increase in surface interaction is lost. With NO as the gas core of the MB, any effect of the charged lipid in promoting interaction is removed. For both NOMBs^+^ (−25%) and NOMBs (−65%), there is an apparent net repulsion of the MBs from a negatively charged surface. 

### 3.2. Assessment of MB Non-Selective Binding Strength in Dynamic Conditions

A large proportion of RAMBs^+^ (>75%) was able to sustain an effective interaction with a negatively charged surface up to wall shear stress values of 0.5 Pa; in contrast, when the charged surface was removed, there was a substantial loss of RAMBs^+^ (>40%) exposed to the same wall shear stress (Figure 3). This attests to the relative strength of electrostatic interaction of RAMBs^+^ with the negatively charged surface. In uncharged RAMBs and RAMB^+^ suspensions, there was no statistically significant difference in their ability to sustain an interaction with either a charged or uncharged surface. Contrary to the findings of static binding experiments, NOMBs^+^ appear better able to maintain an interaction with the artificially charged surface than RAMBs^+^; however, there was no significant difference in between the charged and uncharged NOMBs or RAMBs’ abilities to sustain an interaction with the negatively charged surface under increasing wall shear stress levels (Figure 4). At 1 Pa, >90% of RAMBs irrespective of formulation were flushed out of the system, indicative that the non-selective interaction of room–air MBs with surfaces is non-permanent and easily reversible. It should be noted in contrast that at 1 Pa, 72.9% of NOMBs^+^ were still in contact with the negatively charged surface, compared to 41.4% with the uncharged surface. 

### 3.3. Assessment of MB Interaction with P. aeruginosa Biofilms

Images of biofilms captured with fluorescence microscopy and MBs captured with brightfield and epifluorescence microscopy have been overlaid in ImageJ to provide a comprehensive visual representation of the interaction between MBs and the *P. aeruginosa* biofilm (Figure 5). Visual inspection of the interaction between both cationic MBs^+^ and uncharged MB suspensions with *P. aeruginosa* biofilms and the glass growth substrate indicated that there was a low level of innate MB association to both strata irrespective of net charge. As observed in Figure 5a, clusters of MBs were clearly associated with the glass surface of the growth area, but there was no specific interaction with the distinct areas of biofilm growth. In contrast, DSPC^+^ RAMBs exhibited a significantly (*p* < 0.001) increased quantity of MBs (38%) (Figure 6), which appeared to be more highly localised and non-selectively bound to areas of biofilm (Figure 5b). From visual inspection, there also appeared to be a positive correlation between the quantity of bound MBs and depth of biofilm growth; specifically, in areas where the biofilm is denser, there was a visibly discernible increase in the aggregation of MBs (Figure 5b). 

In this experiment, the innate level of uncharged MB interaction with biofilms and growth surface was on average 10% for uncharged RAMBs and 2% for uncharged NOMBs (Figure 6). Sharing consistency with binding experiments conducted in flow cells with an artificially charged surface (Figure 2), RAMBs^+^ exhibited a significantly (*p* < 0.001) increased level of association to the biofilm (38%) compared to all other formulations; NOMBs^+^ were the second most effective (28%). The incorporation of cationic charge into the NOMBs^+^ shell was shown to significantly increase (*p* < 0.005) their association with the *P. aeruginosa* biofilms and glass substrate, when compared to both uncharged RAMBs and NOMBs. There was an average decrease in non-selective NOMBs^+^ binding of 7% compared to RAMBs^+^, but an average increase in NOMBs^+^ interaction of 16% and 24%, respectively, over uncharged RAMBs and NOMBs (Figure 6). 

## 4. Discussion

Due to the challenges in current research on specific ligand-receptor mediated targeting of bacterial biofilms, it was the aim of this research to validate the use of a non-selective means of targeting the bacterial biofilm. We hypothesised that cationic microbubbles could therefore be utilised as a non-selective means of targeting bacterial biofilms to achieve greater local concentration of MBs associated with and proximal to the biofilm and in turn increase the efficacy of ultrasound-mediated antibiofilm therapy. In order to assess this research aim, a microfluidic flow cell was designed where a glass or quartz surface could be bonded to an acrylic manifold (Figure 1). To provide an effective level of mechanical support to the 170 µm thick glass surfaces used in these experiments (required to allow high magnification microscopy), the manifold selected was a 6 mm thick clear acrylic. This allowed each flow cell to be optically transparent, produced rapidly and precisely by laser cutting, and be cost effective at a material cost of £0.83 per flow cell. Aqueous solutions in contact with a solid surface can create charge on the surface depending on the composition of the solid material and chemical properties of the solution; zeta potential can be used as a measure of the relative charge created at this surface [46]. Borosilicate glass coverslips bonded to the acrylic manifold generate a surface that does not create or sustain a charge, making them a suitable representation of an uncharged surface. Conversely, a quartz surface in contact with PBS at pH 7.4 has been shown to have a zeta potential of approximately −40 mV [47], which provides a valuable artificial control for a surface with a negative electrostatic charge. The artificially anionic and uncharged environments that were created in the flow cell formed an essential proof of concept to determine the ability of cationic MBs to interact with a negatively charged surface. It was important to first characterise this interaction in an environment that is static and can be controlled to best establish if there could be an effect before moving into more complex biological testing on biofilms. This characterisation of interaction focused on inducing contact between RAMBs^+^, RAMBs, NOMBs^+^, and NOMBs and either a negatively charged or uncharged surface. Then, it was studied by inverting the device, creating ideal conditions for weakly associated MBs to passively float away from the surface. As this method is carried out using smooth surfaces with augmented charge, it strictly accounts for MB association induced by electrostatic forces only. This is an important assessment as it provides a baseline for the efficacy of incorporating charge as a means of non-selective targeting without overestimation in the data caused by MB trapping at the surface due to variations in topography intrinsic to biological samples such as biofilms. The zeta potential measurements in this work were carried out on uncharged (DSPC:PEG40s) and cationic (DSPC:PEG40s:DSEPC) MB formulations. As outlined in previous research [40], the measurement was performed on samples that underwent only the first sonication step of the MB production process. Microbubbles are inherently buoyant and will float during measurements in a DLS apparatus, subsequently reducing the accuracy of the measurement. After the first sonication step the sample contains a dispersion of nanoscale lipid vesicles (i.e., microbubble precursors) that are neutrally buoyant, thereby making the measurement possible. Whilst this approach makes the measurement easier to obtain and increases its accuracy, it has been shown that the electrostatic charge of microbubbles and of their precursor lipid vesicles of the same formulation remain consistent [40]. Similarly, the purpose of this measurement was to determine the relative charge created by the MB shell formulation only; therefore, the sonication step was only performed in air. The DSPC:PEG40s:DSEPC cationic MB formulation was shown to have a zeta potential of 19.74 ± 0.95 mV (Table 3), which is congruent with other research that has produced cationic MBs with a zeta potential of approximately 25 mV using similar sonication method, lipid composition, and molar ratios employed in this work [40]. An indicative assessment of cationic MB charge has been carried out elsewhere [41]; this work showed that the most frequently reported range of MB charge was 15 mV to 40 mV; however, a higher reported charge of 60 mV was also previously determined for a DSTAP:DSPC:PEG40s lipid-shelled MB used in gene delivery to skeletal muscle. 

The data for static interaction of RAMBs^+^ provided an insight into the efficacy of charge as a means of non-selective binding; there was a significant increase of >40% in RAMBs^+^ associated with the negatively charged surface compared to the uncharged surface, which affirms that the observed effect is due to interaction of the cationic MB and the anionic surface (Figure 2). Uncharged RAMBs displayed no significant percentage increase in MBs associated with the charged surface compared to the uncharged surface. This provides evidence that the non-selective binding effect is explicitly caused by the incorporation of DSEPC in the microbubble shell. Given this evidence that cationic MBs containing DSEPC promote electrostatic interactions with anionic surfaces, it appears anomalous that there was a greater loss of NOMBs^+^ from the negatively charged surface compared to the uncharged surface. It was hypothesised that a NO-specific mediated interaction was causing interference with the electrostatic interactions, consequently reducing the binding efficacy of the charged MBs. Evidence for this effect is clear in the data for the interaction of uncharged NOMBs with negatively charged surfaces; there was a highly statistically significant difference between the number of NOMBs^+^ lost from the negatively charged surface compared to the uncharged surface. It would therefore appear that the presence of DSEPC effectively mitigates the repulsive effect seen in MBs with an NO core from anionic surfaces, since 40% more DSPC NOMBs^+^ remained in contact with the charged surface than DSPC NOMBs (Figure 2). In all experiments carried out in this research involving NOMBs and negatively charged surfaces, there has been a demonstrably weaker attraction between NOMB and negatively charged substrates compared to RAMBs, despite the incorporation of a cationic moiety into the NOMB shell. Although NO is a highly reactive molecule due to the presence of an unpaired electron, it is not an anionic molecule in either the gas or solution phase [48]. However, under physiological and environmental conditions, the bioactivity of NO is limited by its oxidation to the stable anionic products nitrite (NO^2−^) and nitrate (NO^3−^), with nitrate being the prevailing oxidation product formed [49]. As demonstrated in other research, it is highly likely that the repulsion of NOMBs from negatively charged surfaces seen in these experiments is caused by the accumulation of the anionic products NO^2−^ and NO^3−^ in solution [50,51,52]. The exception to this effect seen in experiments conducted under flow to assess NOMB binding strength (Figure 4) is likely due to the flow of PBS facilitating removal of oxidised products of NO, thereby reducing the concentration of negative ions in the fluid and mitigating their interference with the MB binding process. Building on this assumption, it would seem plausible that the loss of NO from NOMBs by oxidation would render their utility as a means of inducing dispersal in biofilms invalidated. However, in delivering a solution that invariably contains the oxidative products of NO, we provide bacterial cells within the biofilm with an important source of nitrate that can be recycled to form NO via bacterial nitrate reductases [53]. 

Research in this area has explored the antimicrobial effects of nitrite in solution [49]; this has been demonstrated in cystic fibrosis models of *P. aeruginosa* infection, where a 3-log reduction in bacteria was seen after a four-day prolonged exposure to 15 mM nitrite [54]. Further research in vitro has shown that the conversion of nitrite to NO is potentiated in the presence of ascorbic acid, which is produced by clinically important species of bacteria such as *P. aeruginosa* and *S. aureus* and results in enhance bactericidal activity [55,56]. Importantly, the impact of providing bacterial biofilms with sources NO^3−^ that can be reduced endogenously to NO by aerobically grown *P. aeruginosa* biofilms has been characterised by Rodgers et al. [57], who have shown that low concentrations (100 nM to 1 μM) of endogenous NO can induce biofilm dispersal. The presence of nitrite and nitrate in the effluent of biofilms usually only occurs in mature biofilms (i.e., >5 days old), at which point the endogenous production of NO induces dispersal of the biofilm architecture [57]. Therefore, by providing biofilms with an abundant source of nitrite and nitrate in the NOMB solution, there will be a corresponding upregulation in denitrification and endogenous production of NO to more rapidly induce a major biofilm dispersal event. The expedited endogenous production of NO in biofilms for dispersal may prove to be an interesting avenue of exploration for future work in this area. The relative strength of the electrostatic interaction between cationic MBs and an anionic surface was assessed by exposing MBs in the flow cell to increasing levels of wall shear stress from 0.1 to 1 Pa by altering the volumetric flow rate through the channel (Table 2). For physiological reference, mean wall shear stress in large veins is typically <0.1 Pa, whilst it is higher in arterioles at up to 8 Pa [58]. Previous research has demonstrated that wall shear stress values up to 0.06 Pa could promote microbubble binding; however, there is a decline in attachment after this point [25]. We hypothesised that in a similar manner, we would observe a decline in surface-associated MBs in response to increasing wall shear stress levels. It was confirmed that the dissociation of MBs from both charged and uncharged surfaces was less pronounced at wall shear stress values up to 0.5 Pa, but occurred more rapidly from 0.5 Pa to 1 Pa for all formulations (Figure 3). Phospholipid-coated ανβ3-targeted MBs used in endothelial cell targeting by Langeveld et al. remained attached to their target under flow up to shear stress values of approximately 0.2 Pa [59]. There was a significant reduction in the number of bound MBs from approximately 0.5 Pa, which is consistent with levels of attachment and loss of cationic MBs under flow in this work. Interestingly, Langeveld et al. also compared how the homogeneity of the ligand distribution affected MB binding. They showed that MBs with a more homogenous ligand distribution had a higher binding efficacy than those with a heterogenous distribution [59]; future work in this area could investigate how the arrangement of lipids in a cationic MB may also affect MB binding efficacy. In our investigation, both RAMBs^+^ and NOMBs^+^ were able to remain in contact or continue accumulating on the negatively charged surface up to wall shear stress values of 0.2 Pa and 0.5 Pa, respectively. This result would suggest that in comparison to studies that have assessed the binding strength of ligand-targeted MBs under increasing wall shear stress levels, the electrostatic interaction induced here can withstand a greater level of wall shear stress before initial detachment begins [27,60]. These findings are directly concurrent with work carried out by Edgeworth et al., who demonstrated that MBs electrostatically attached to a surface remained bound up to wall shear stress values of 0.66 Pa [61]. However, in the present study, RAMBs^+^ and NOMBs^+^ were reduced by approximately 90% and 30%, respectively, of their maximum coverage surface at a wall shear stress of 1 Pa (Figure 3 and Figure 4); whereas, ligand-receptor targeted MBs have been shown to withstand wall shear stress levels of up to 50 Pa [61]. It is important to note, however, that this may not be an accurate representation of the MB binding and detachment profiles that would occur in vivo. As most studies can only estimate the local site density of the target receptor, there may be a significant over- or under-estimation in the data [27]. 

Though the microfluidic flow cells provided a valid and essential initial assessment of cationic MB interactions with charged surfaces to be able to understand the interaction between cationic MBs and biofilms more comprehensively, they were not fit for purpose. Due to the difficulty in analysing biofilm architecture and MBs simultaneously and a lack of translational applicability in terms of how MB suspensions would be applied clinically, biofilms were instead grown in Ibidi^®^ dishes that feature a 170 µm thick glass coverslip growth area. This provided a means to grow biofilms on a surface where MB suspensions could be applied topically under clinically relevant conditions. The data present the percentage area occupied by MBs in microscopy images (Figure 6) which were retained after a three-step washing process to remove all weakly associated MBs. These data were averaged across three independent biofilms to demonstrate consistency in MB binding regardless of biofilm topography. The data shared a strong correlation with the effects observed in flow cell experiments; RAMBs^+^ exhibited a significantly (*p* < 0.001) increased level of non-selective association with the biofilm (38%) compared to all other formulations, with NOMBs^+^ being the second most effective (28%) (Figure 6). Caudwell et al. showed that the *S. aureus* specific binding Affimer AClfA1 conjugated with MBs, could achieve a nine-fold increase in MBs bound to the biofilm over non-targeted control MBs [23]. The authors note that utilising MBs in biofilm targeting has some limitations; principally, MBs are inherently unstable over time and can be destroyed during injection/administration processes [23]. Moreover, though there is great value in the specificity of the ligand for targeting biofilms, a potential limitation of this interaction is that this specificity may be highly variable between strains of the same bacterial species. However, this work sought to overcome such a limitation by utilising the non-selective nature of cationic charge to promote MB–biofilm interaction. In this work, the MB coverage as a proportion of the total area in the image was assessed, as opposed to limiting the analysis to only the percentage area of the biofilm covered by MBs. Though this includes MBs non-selectively interacting with the glass surface of the growth area, it provides a more accurate representation of what is likely to occur in vitro. By not eliminating MBs that were peripheral to the biofilm but not directly associated, we gain a better understanding of the innate level of MB retention due to varied surface topography. Moreover, even though they are not directly associated with the biofilm, proximal MBs may still have a significant impact on the overall efficacy of treatment due to the cavitation regimes induced upon exposure to ultrasound [13,62,63]. Specifically, fluid flows induced by microbubble cavitation (also known as microstreaming) may potentiate local drug delivery by imparting greater shear stress on the biofilm to increase permeability and enhance transport of chemical species [64,65]. As demonstrated by Pereno et al. [65], microstreaming velocities can peak in the order of 1000 µm/s and be maintained at 100 µm/s at distances > 1 mm from the source MBs cavitating in response to a 1 MHz ultrasound stimulus. It is important to note that the reduced number of surface-associated NOMBs in this experiment provides further support to the hypothesis developed from flow cell experiments; there is a net repulsion between MBs with a NO core and surfaces with a net negative charge, i.e., biofilms in this experiment. Evidence to support this can be seen in the significant increase (*p* = <0.0001) in NOMB^+^ binding to biofilms compared to their uncharged counterparts; it can thus be hypothesised that the positive charge of the lipid in this instance is aiding the MB in overcoming the effect of NOMB repulsion from the biofilm. There is still a significant amount of NOMBs^+^ bound to biofilms when compared to uncharged RAMBs (*p* = <0.0005) and NOMBs (*p* = <0.0001), which only cover a maximum surface area of 10% and 2%, respectively (Figure 6). These data, coupled with the significant difference (*p* = <0.05) between NOMBs^+^ and RAMBs^+^, add credence to the claim that the cause of this apparent repulsion is NO or more specifically its anionic oxidised products. 

## 5. Conclusions

It was the aim of this research to determine if cationic MBs could be used as a means of non-selectively targeting the bacterial biofilm to achieve a greater local concentration of MBs associated with and proximal to the biofilm and in turn increase the efficacy of ultrasound-mediated antibiofilm therapy. The research aims were achieved by assessing the binding affinity and strength of cationic MBs to uncharged and charged surfaces under static and dynamic conditions within a flow cell capable of augmenting surface charge that was developed in this research to facilitate the study of these interactions in a controlled manner. Whereby, the addition of a cationic charge to the MB shell facilitated a consistent increase in the number of MBs that could bind to a negatively charged substrate. This interaction with a negatively charged substrate was shown to be sustained up to wall shear stress levels of 1 Pa, with cationic MBs remaining in consistently greater numbers compared to their uncharged counterparts. Subsequent testing demonstrated the efficacy of cationic MBs in significantly enhancing MB contact when applied directly to a *P. aeruginosa* biofilm. This corresponded to a significant 26% and 24% increase in the number of RAMBs^+^ and NOMBs^+^, respectively, able to bind and sustain contact with the biofilm. This work demonstrated for the first time that cationic MBs have the ability to increase proximity, promote, and sustain contact with bacterial biofilms, validating their utility as a means of non-selectively targeting the biofilm. 

## Figures and Tables

**Figure 1 pharmaceutics-15-01495-f001:**
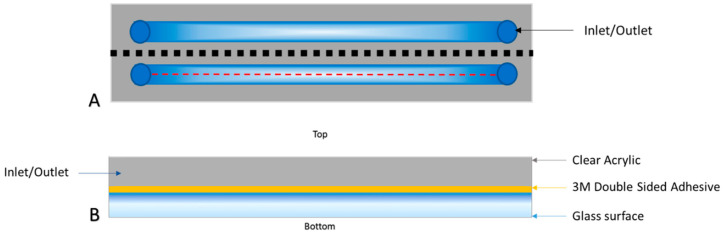
Schematic diagram of microfluidic flow cell with main features highlighted. The flow cell measures 75 × 25 mm (length × width), and the fluid channel is 0.14 mm high and 3.5 mm wide. (**A**) Top view of the flow cell with a dashed red line indicating the centreline of the channel, which is approximately 1.7 mm equidistant from the lateral walls of the channel. (**B**) Cross-sectional view of **(A)** taken at the dotted black line, with ‘top’ and ‘bottom’ orientation indications (figure not to scale).

**Figure 2 pharmaceutics-15-01495-f002:**
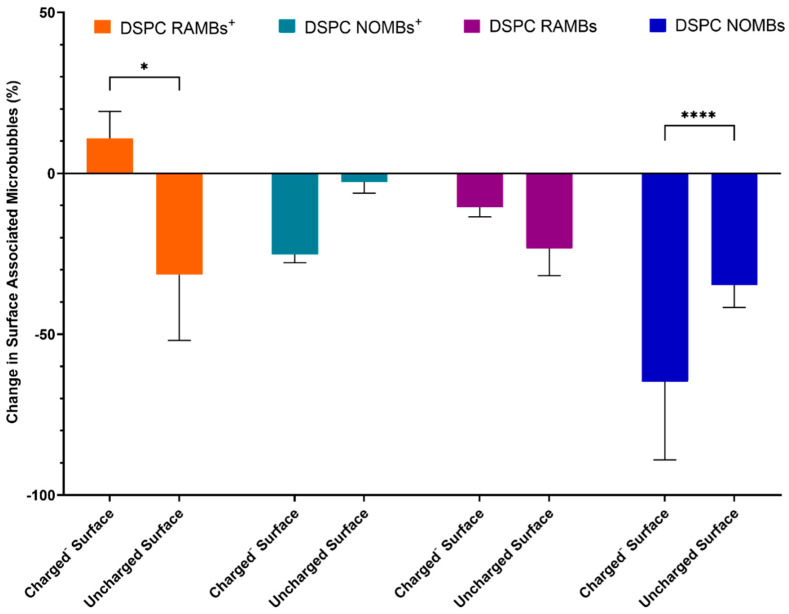
Assessment of the change in surface-associated cationic RAMBs^+^ and NOMBs^+^ and their respective uncharged equivalents after a 60 s interaction with either a negatively charged or uncharged surface. The data show that the inclusion of the positively charged DSEPC lipid in the shell of DSPC RAMB^+^, significantly increased the quantity of MBs that can maintain contact with a negatively charged surface by 40%. For both NOMB and NOMB^+^ suspensions, there was a consistent net loss of MBs that interacted with either surface, but this loss increased significantly in the presence of a negatively charged surface. Approximately 65% of NOMBs failed to remain in contact with the negatively charged surface, compared to a loss of 40% from the uncharged surface. This apparent repulsion of NOMBs from the negatively charged surface is mitigated somewhat by the presence of the cationic DSEPC lipid in NOMBs^+^, of which only 25% were lost from the negatively charged surface and <10% were lost from an uncharged surface. Error bars represent the standard deviation of the mean. * = *p* < 0.05, and **** = *p* < 0.0001.

**Figure 3 pharmaceutics-15-01495-f003:**
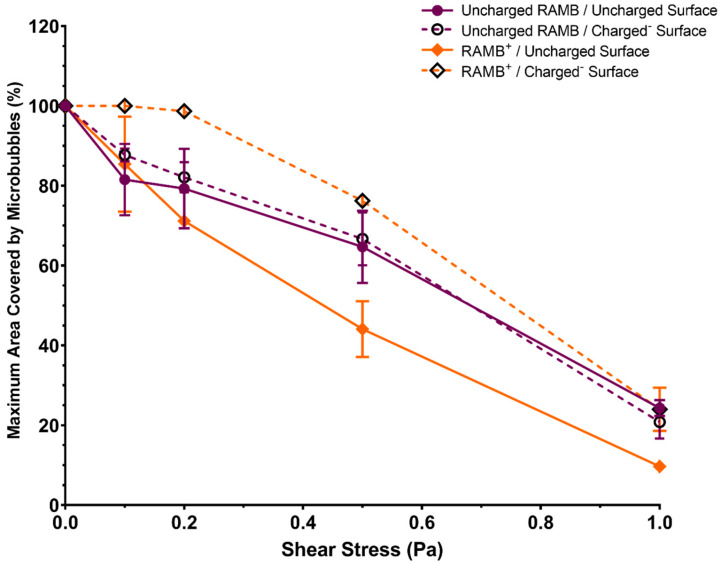
Percentage of maximum area covered by MBs for uncharged RAMB (9:0.5 DSPC:PEG40s) and cationic RAMB^+^ (9:0.5:1 DSPC:PEG40s:DSEPC). The total area covered by each MB composition was assessed over 10 images, captured after the application of incremental wall shear stress values from 0 to 1 Pa. Solid lines represent MB association with the uncharged surface; dashed lines represent MB association with the negatively charged surface. RAMBs^+^ demonstrated the highest affinity for the negatively charged surface, with no substantial loss of MBs in contact with the surface until wall shear stress values > 0.2 Pa. All values were normalised by taking the percentage value of the area covered by MBs at 0 Pa as the baseline maximum percentage area covered (100%). Error bars represent the standard deviation of the mean.

**Figure 4 pharmaceutics-15-01495-f004:**
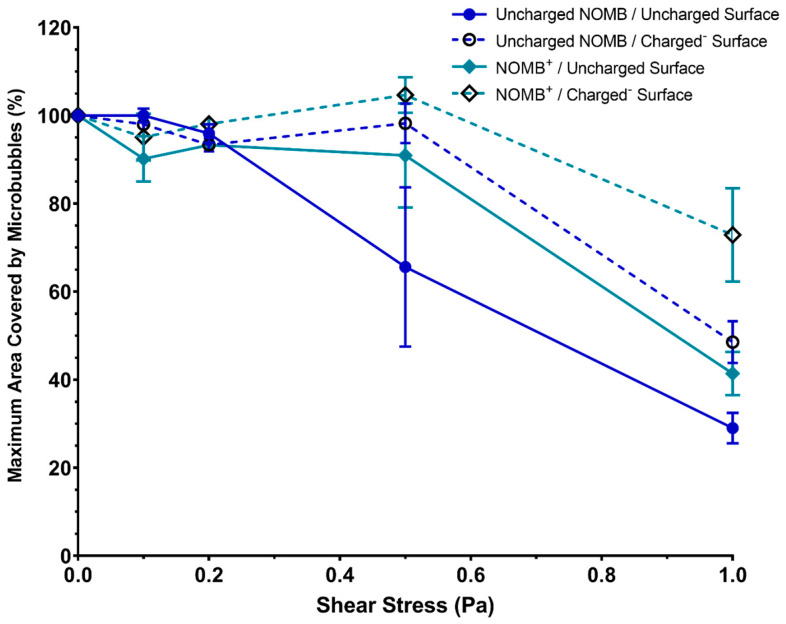
Percentage of maximum area covered by MBs for uncharged NOMBs (9:0.5 DSPC:PEG40s) and cationic NOMB^+^ (9:0.5:1 DSPC:PEG40s:DSEPC). The total area covered by each MB composition was assessed over 10 images, captured after the application of incremental wall shear stress values from 0 to 1 Pa. Solid lines represent MB association with the uncharged surface; dashed lines represent MB association with the negatively charged surface. Uncharged NOMBs interacted with the uncharged surface as expected, with a rapid decline in MBs associated with the surface in response to wall shear stress > 0.2 Pa. For both NOMB and NOMB^+^ suspensions, there was an accumulation of MBs on the charged surface up to 0.5 Pa, with only 50% and 30% loss respectively in maximum coverage at 1 Pa. All values were normalised by taking the percentage value of the area covered by MBs at 0 Pa as the baseline maximum percentage area covered (100%). Error bars represent the standard deviation of the mean.

**Figure 5 pharmaceutics-15-01495-f005:**
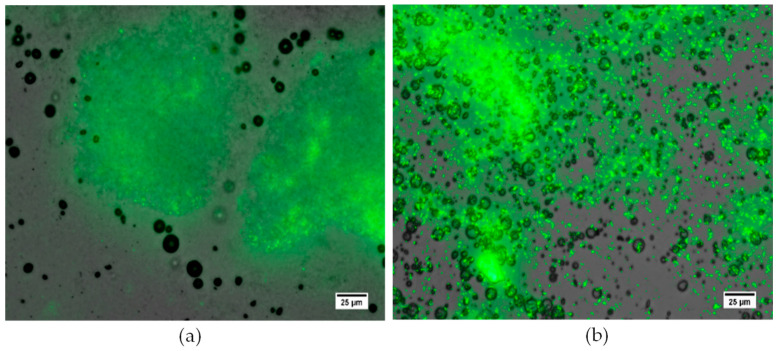
Visual assessment of the interaction between uncharged DSPC RAMBs (**a**) and cationic DSPC RAMBs^+^ (**b**) with a *P. aeruginosa* biofilm (stained with Syto9, green). Visual inspection confirms, there is an increased level of non-selective binding of cationic DSPC RAMBs^+^ to defined areas of biofilm growth, compared to a low level of residual uncharged MB interaction with the glass growth surface and little to no association of uncharged MBs to defined areas of biofilm growth. Scale bar = 25 µm.

**Figure 6 pharmaceutics-15-01495-f006:**
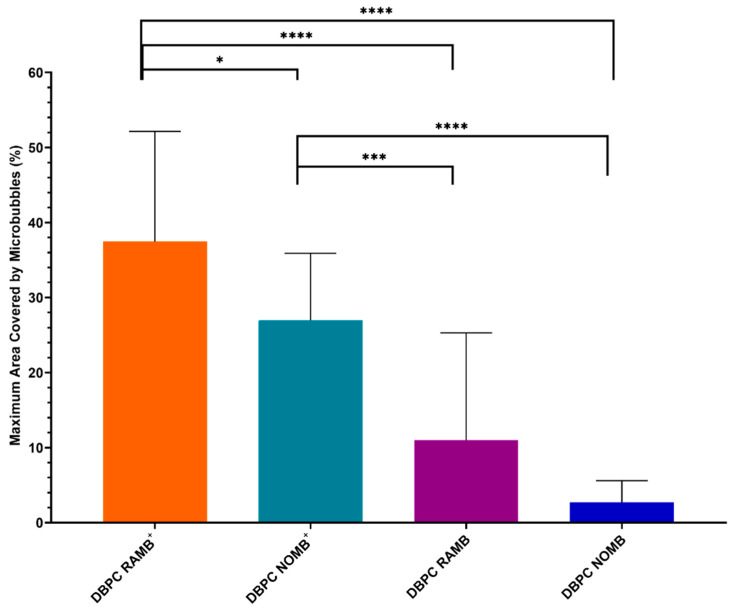
The maximum average percentage area covered by each MB formulation, assessed by analysing three independent biofilm samples for each of the four MB formulations tested. After administration of MBs to biofilms and subsequent washing to remove all MBs with a weak or no association to the biofilms, the average percentage coverage of the remaining MBs was 37% for RAMBs^+^, 27% for NOMBs^+^, 11% for uncharged RAMBs, and 3% for uncharged NOMB suspensions; error bars report the standard deviation of the mean. Quantification of MB coverage was assessed by determining the percentage area that MB formulations covered in each image, which incorporated both MB association to the *P. aeruginosa* biofilms and glass substrate of the Ibidi^®^ dish growth area. * = *p* < 0.05, *** = *p* < 0.0005, and **** = *p* < 0.0001.

**Table 1 pharmaceutics-15-01495-t001:** Stock concentrations, molar ratio compositions, and volumes used for neutral and cationic MBs.

MB Composition	DSPC	PEG40s	DSEPC
Stock concentrations (mg/mL)	25	10	10
Molar ratio			
Uncharged	9	0.5	0
Cationic (+)	9	0.5	1
Preparation volumes (µL) to produce 2 mL of MB suspension with a final lipid concentration of 4 mg/mL			
Uncharged	210	75	0
Cationic (+)	190	68	57

**Table 2 pharmaceutics-15-01495-t002:** Volumetric flow rates of PBS corresponding to mean wall shear stress values to be exerted on MBs bound to the glass surface. The volumetric flow rates for each required mean wall shear stress were calculated from the pressure drop along the channel using Equations (2) and (3).

Mean Wall Shear Stress (Pa)	0.1	0.2	0.5	1
Pressure drop along channel (Pa)	97.31	194.63	486.57	973.14
Volumetric flow rate (mL/min)	0.06	0.13	0.32	0.65

**Table 3 pharmaceutics-15-01495-t003:** The mean (n = 3) diameter with standard deviation, mean concentration, and mean zeta potential with standard deviation of each MB formulation immediately after production.

Microbubble Formulation	Mean MB Diameter (µm)	Mean MB Concentration (MB/mL)	Zeta Potential (mV)
Uncharged RAMB	3.48 ± 2.28	5.63 × 10^7^	−2.44 ± 2.60
Uncharged NOMB	2.85 ± 2.72	1.71 × 10^8^	
Cationic RAMB	5.25 ± 2.70	1.96 × 10^8^	19.74 ± 0.95
Cationic NOMB	3.22 ± 1.74	2.69 × 10^8^	

## Data Availability

Data can be made available upon request.

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
