# Peer review of "Cationic Microbubbles for Non-Selective Binding of Cavitation Nuclei to Bacterial Biofilms"

_pharmaceutics, 2023, doi:10.3390/pharmaceutics15051495_

Round 1
Reviewer 1 Report
This is a really interesting study in which the authors generated microbubbles that were either cationic or neutral in charge characterization. These bubbles were then used to target preformed Pseudomonas aeruginosa biofilms and the efficacy of this treatment was investigated using a viability stain (Live/Dead) and a flow cell culture apparatus. The authors' manuscript currently adds to prior work being done by a number of investigators. I have several questions that I would like the authors to address:
1) Microbubble size was determined by analysis of images primarily using Image J. However in looking at the size distribution in Figure A1, there appears to be a considerable variation of size. I have several questions related to this: A) what is the size distribution, B) was there any correlation with the chemical characteristics of the microbubble and size distribution, C) how did the relative sizes compare as to their efficacy against biofilms, and D) do the authors have any indication of the depth of penetration of the various microbubbles into the biofilms?
2) In a related question, what residual material if any might remain following biofilm treatment?
3) Is there any variability in the NO concentration within the microbubbles?
4) Is there any change in efficacy of this treatment when biofilms are grown in different nutritional conditions?
Author Response
1) Microbubble size was determined by analysis of images primarily using Image J. However in looking at the size distribution in Figure A1, there appears to be a considerable variation of size. I have several questions related to this:
A) What is the size distribution
Graphs have been provided for information on the microbubble size distribution and concentration observed over 60 minutes at room temperature for each formulation, which we have added to the supplementary data of the paper.
B) was there any correlation with the chemical characteristics of the microbubble and size distribution
Cationic RAMBs are larger than their uncharged counterparts after production, with a larger standard deviation in the diameter of the population. Their increase in diameter over time, however is comparable to uncharged RAMBs and their concentration drops less severely. The incorporation of a cationic charge does not seem to impact post-production size or concentration of NOMBs; however, it is important to note that in both instances there is no statistically significant difference in size or concentration observed between any of the formulations.
C) how did the relative sizes compare as to their efficacy against biofilms
In our other published work (https://www.frontiersin.org/articles/10.3389/fcimb.2022.956808/full) on the application of these formulations, it was shown that cationic microbubbles required higher driving acoustic pressures, in order to obtain a comparable acoustic response to their uncharged analogues (confirmed by high-speed imaging, supplementary materials).
D) do the authors have any indication of the depth of penetration of the various microbubbles into the biofilms?
The depth of the biofilms tested naturally varied, however the minimum biofilm depth was typically 20 µm and in some areas commonly up to 50 µm. In our paper that focused on ultrasound stimulation of these MBs we saw clear evidence that the MBs penetrated the full depth of these biofilms, resulting in high levels of biofilm clearance from the surface (with and without nitric oxide). We can summarise from this information that in response to ultrasound stimulation the MBs penetrate the biofilm substantially. When applied to the biofilm and in the absence of ultrasound, it was only possible to determine that the MBs interacted with the most superficial (exposed) portions of biofilm and free-floating bacteria.
2) In a related question, what residual material if any might remain following biofilm treatment?
In our analysis of the application of these formulations for biofilm dispersal (https://www.frontiersin.org/articles/10.3389/fcimb.2022.956808/full) the biofilm material remaining was variable between each different formulation (as published), but results were consistent between biological replicates of the same formulation. This earlier publication specifically explored the relationship between dispersal of biofilms and quantification of bactericidal activity, as we are acutely aware that we would not want to disperse viable bacteria. In terms of any MB shell constituent material remaining after treatment, videos of the MB response captured with high-speed microscopy shows that a proportion of microbubbles are destroyed. Though the intended application discussed here is topical, clinically approved formulations of MBs are shown to be safely excreted after intravascular administration so we foresee no negative impacts of residual material following treatment.
3) Is there any variability in the NO concentration within the microbubbles?
In our previous paper (https://www.frontiersin.org/articles/10.3389/fcimb.2022.956808/full) reporting on the application of these formulations for biofilm dispersal, the estimated average (N = 3) NO concentration within the MB suspension was 16.5 µM (SD ± 1.45 µM). Though there is variability, the concentration is maintained at therapeutically effective levels.
4) Is there any change in efficacy of this treatment when biofilms are grown in different nutritional conditions?
In exploring the use of MBs for treating biofilms, we aimed to replicate only the most biologically applicable conditions for wound biofilms. In this we developed a modified nutrient-rich wound constituent medium, based on the Lubbock wound model, and we included a fibronectin coating on the growth surface. We ensured biofilms possessed a dense extracellular matrix and were robustly attached, effectively replicating as best as we can in vitro the worst-case scenario for an in vivo biofilm. Therefore, the only avenue to explore here would be less nutrient-rich environments in which biofilms can grow, but this was outside the scope of the present work. Based on the work of other groups we can assume that with poorer nutritional conditions, as shown when biofilms are grown in medium such as M9, biofilms are typically much less dense with a more transient attachment to the growth surface.
Reviewer 2 Report
The manuscript reports about cationic microbubbles, designed for non-selective binding to bacterial biofilms. The authors characterized the effect of a positive charge of MBs on their binding to neutral and negative surfaces in both static and dynamic conditions. They also analyzed the interaction of MBs with P. aeruginosa biofilm.
The topic is interesting, the manuscript is well-organized and the presentation of the results is accurate. I have some comments that could be faced to the manuscript before publication.
1. In Table 3, authors report only two values of Zeta potential for the four sample analyzed: I suggest to report all the measured values. This could also help to improve the interpretation of the differences observed between NOMBs and RAMBs in the binding experiments in static conditions (see comment 2).
2. I am not convinced by the realization of binding experiments in static conditions, please clarify the following points:
- in experiments, the images before and after interaction are acquired with the binding surface placed to be the bottom of the flow cell: in this way the measure could be affected by the presence of debris and particles that are not gas-filled (not MBs) that sink to the bottom of the cell;
- authors use the percent variation of the number of MBs associated to the surface as the main observable to characterize the binding. In this way, if some MBs are already bound to the surface in the images "before interaction" (possibly due to strong electrostatic attraction during injections even if the binding surface is placed on the bottom of the cell?) would result in lowered value of the measured percentage and therefore would be interpreted as lower binding efficacy;
- alternative experimental strategies - such as imaging the binding surface when placed on the top, immediately after reversing the cell to exclude the MBs that are floating up - and the use of a different observable - e.g. the difference between the numbers of associated MBs before and after interaction - could be considered.
3. The achieved binding results should be discussed in comparison to other researches analyzing the binding of MBs mediated by specific interactions (see, for example 10.1016/j.ultrasmedbio.2019.09.011, 10.1016/j.bioflm.2022.100074, 10.1016/j.jcis.2020.06.009, 10.1039/d0bm01857k, 10.1186/1477-3155-12-24 and 10.3390/pharmaceutics14020311)
Author Response
1. In Table 3, authors report only two values of Zeta potential for the four sample analyzed: I suggest to report all the measured values. This could also help to improve the interpretation of the differences observed between NOMBs and RAMBs in the binding experiments in static conditions (see comment 2).
Two zeta-potential measurements are reported as these corresponded to the two types of microbubble shell constituents analysed, specifically DSPC:PEG40s and DSPC:PEG40s:DSEPC. As noted in the paper and in other literature, zeta-potential measurements using a DLS apparatus are not performed on MBs directly, but rather on the shell constituents alone, due to limitations of the instrumentation. As such, there is no gas core to demonstrate the effect that the gas formulation would have on the charge of the produced MBs.
2. I am not convinced by the realization of binding experiments in static conditions, please clarify the following points:
- in experiments, the images before and after interaction are acquired with the binding surface placed to be the bottom of the flow cell: in this way the measure could be affected by the presence of debris and particles that are not gas-filled (not MBs) that sink to the bottom of the cell;
The images are captured after the prescribed period of MB association with the charged/uncharged surface; after reorientation of the device to place this surface downwards for imaging, we do not believe there is sufficient time for most debris to settle back to this surface and occlude images. Moreover, should this occur it would be mitigated in the image processing steps themselves, as the analysis is limited to only counting defined highly circular objects. It is worth noting that other publications assessing the binding affinity of functionalised MBs and biofilms have employed the same method of inversion for inducing contact before reorientation for imaging https://www.sciencedirect.com/science/article/pii/S2590207522000089.
- authors use the percent variation of the number of MBs associated to the surface as the main observable to characterize the binding. In this way, if some MBs are already bound to the surface in the images "before interaction" (possibly due to strong electrostatic attraction during injections even if the binding surface is placed on the bottom of the cell?) would result in lowered value of the measured percentage and therefore would be interpreted as lower binding efficacy;
In these static experiments each MB suspension was allowed to interact with the charged or uncharged surface for 60 seconds; before ‘post interaction’ images were then captured. A 60 second interaction period was selected as also aligns with the time taken for the majority of the MBs in suspension to passively float towards the surface. In this way the ‘before’ images set the baseline for attachment at 100%, and the inversion of the flow cell allows all weakly attached MBs to passively float away from the surface. The subsequent change in the number of MBs, i.e. the number remaining on the surface once reoriented, was determined by comparing the number of MBs attached to the surface ‘before’ to ‘after’. Therefore, by design there were MBs present in the ‘before’ images as this established the baseline for attachment, including the inevitable underlying level of attachment of MBs due to surface tension and device geometry independent of charge. This aspect of the work was the proof of concept for later work, simply to show that a cationic charge would allow more MBs to stick to a negatively charged surface than a MB without a charge.
- alternative experimental strategies - such as imaging the binding surface when placed on the top, immediately after reversing the cell to exclude the MBs that are floating up - and the use of a different observable - e.g. the difference between the numbers of associated MBs before and after interaction - could be considered.
The alternative method suggested here was not feasible with the equipment available to the authors when this work was carried out; there were only inverted microscopes available. Therefore, interaction had to be assessed by allowing MBs to make contact with the surface and then assess how many were removed after the glass surface was reoriented to face downwards. Any MBs weakly associated with the surface during this inversion process would have detached and floated away from the surface, so we don’t believe there would be any significant impact on the data from an overestimation of the number of MBs at the beginning of the experiment.
3. The achieved binding results should be discussed in comparison to other researches analyzing the binding of MBs mediated by specific interactions (see, for example 10.1016/j.ultrasmedbio.2019.09.011, 10.1016/j.bioflm.2022.100074, 10.1016/j.jcis.2020.06.009, 10.1039/d0bm01857k, 10.1186/1477-3155-12-24 and 10.3390/pharmaceutics14020311)
We have included references to the publications listed here where appropriate in the work; including comparisons between the present study and these publications when feasible.
Reviewer 3 Report
The current work showed that cationic MBs could increase proximity, targeting bacterial biofilms. The study is well-designed.
Minors
- The authors have to elucidate the importance of bacterial biofilm control in the introduction.
- The used P. aurginosa was clinically isolated or it has ATCC and if it is strong biofilm forming or not, and should be written in italic, please revise
No major linguistic mistakes
Author Response
The current work showed that cationic MBs could increase proximity, targeting bacterial biofilms. The study is well-designed.
- The authors have to elucidate the importance of bacterial biofilm control in the introduction.
A section better highlighting the importance of controlling biofilms to frame the context of this work has been added to the Introduction, using tracked changes.
- The used P. aurginosa was clinically isolated or it has ATCC and if it is strong biofilm forming or not, and should be written in italic, please revise
This was an oversight on our part; the clinical isolate PAO1 was used, this information has now been added to the manuscript. This is widely known to be an excellent biofilm forming strain, commonly used in research of this nature. To the best of our knowledge, the species name is written in italics throughout the manuscript.
Reviewer 4 Report
The authors showed for the first time that microbubbles whose wall contains cationic phospholipid 1,2-132 Distearoyl-sn-Glycero-3-ethylphosphocholine (DSEPC) - i.e. cationic microbubbles - can effectively bind to negatively charged surface elements of the bacterial biofilm - non-specific electrostatic interaction. In this way, the therapeutic gas is delivered directly to the target surface, which is important for unstable gases like NO.
Experimental protocols are written in detail. Results and discussion are in adequate mutual correlation.
Minor:
1. How does it affect the stability of cationic microbubbles if the amount of DSEPC varies during their synthesis, how does the size of microbubbles with cationic DSEPC change compared to neutral microbubbles, does increasing the amount of DSEPC increase electrostatic repulsive interactions on the surface of microbubbles, which leads to their disintegration, or during their synthesis (fabrication) their size increases so that the surface charge density is always constant?
2. In the real application of cationic microbubbles, are there negatively charged molecules in biological systems that can bind to them and neutralize their positive charge (before they reach the negatively charged bacterial biofilm)?
3. The preparation of microbubbles is written in detail, to make it easier to follow the description, the authors could also show it schematically.
Author Response
1. How does it affect the stability of cationic microbubbles if the amount of DSEPC varies during their synthesis.
We agree with the reviewer that this would be an interesting subject of investigation. In this study, the molar ratio between DSPC, DSEPC and PEG40s was selected in such a way to achieve efficient binding to the biofilm surface and minimise costs. Further studies could evaluate the effect of varying the molar amount of DSEPC on the microbubble’s stability and binding strength.
how does the size of microbubbles with cationic DSEPC change compared to neutral microbubbles.
As shown in the paper cationic formulations are larger than their neutral counterparts, but due to the method of production resulting in a broad size population regardless of shell composition, this difference is non-significant.
does increasing the amount of DSEPC increase electrostatic repulsive interactions on the surface of microbubbles, which leads to their disintegration, or during their synthesis (fabrication) their size increases so that the surface charge density is always constant?
We haven’t observed evidence of MB disintegration in cationic microbubbles that may be specifically attributed to their charge. We agree that repulsive forces between lipid molecules may be responsible for changes in MB diameter in charged microbubbles compared to their neutral counterpart. However, this mechanistic analysis was beyond the scope of the present study.
2. In the real application of cationic microbubbles, are there negatively charged molecules in biological systems that can bind to them and neutralize their positive charge (before they reach the negatively charged bacterial biofilm)?
This is an excellent point and one of the principal reasons this is a non-selective means of targeting the biofilm; constituents of the wound bed such as epithelial cells, red blood cells and collagen are also negatively charged. However, we do not see this as a limitation in this application, as the more MBs we can attach to the wound, biofilm, and surrounding constituents the greater opportunity for enhanced drug delivery at the biofilm and wound site there is. We believe that this could pose a greater challenge for applications requiring intravenous administration.
3. The preparation of microbubbles is written in detail, to make it easier to follow the description, the authors could also show it schematically.
We thank the reviewer for their suggestion. We believe the method is comprehensively described and provides the means of directly replicating it. A number of publications pertaining to the production of MBs have been cited throughout the work, some of which do contain visual representations already.